# Cryo-EM structures of the pore-forming A subunit from the *Yersinia entomophaga* ABC toxin

Sarah J Piper [1,2], Lou Brillault[1,2], Rosalba Rothnagel[2], Tristan I Croll[3], Joseph K Box[1], Irene Chassagnon[1], Sebastian Scherer[4], Kenneth N Goldie[4], Sandra A Jones[5], Femke Schepers[6,7], Lauren Hartley-Tassell[8], Thomas Ve [8], Jason N Busby[9], Julie E Dalziel [7], J Shaun Lott [9], Ben Hankamer[2], Henning Stahlberg [4], Mark R H Hurst[5] & Michael J Landsberg [1,2]

ABC toxins are pore-forming virulence factors produced by pathogenic bacteria. YenTcA is the pore-forming and membrane binding A subunit of the ABC toxin YenTc, produced by the insect pathogen *Yersinia entomophaga*. Here we present cryo-EM structures of YenTcA, purified from the native source. The soluble pre-pore structure, determined at an average resolution of 4.4 Å, reveals a pentameric assembly that in contrast to other characterised ABC toxins is formed by two TcA-like proteins (YenA1 and YenA2) and decorated by two endochitinases (Chi1 and Chi2). We also identify conformational changes that accompany membrane pore formation by visualising YenTcA inserted into liposomes. A clear outward rotation of the Chi1 subunits allows for access of the protruding translocation pore to the membrane. Our results highlight structural and functional diversity within the ABC toxin subfamily, explaining how different ABC toxins are capable of recognising diverse hosts.

[1] School of Chemistry and Molecular Biosciences, The University of Queensland, St Lucia Queensland 4072, Australia. [2] Institute for Molecular Bioscience, The University of Queensland, St Lucia Queensland 4072, Australia. [3] Cambridge Institute of Medical Research, University of Cambridge, Cambridge Cambridgeshire CB2 0XY, United Kingdom. [4] Centre for Cellular Imaging and NanoAnalytics, Biozentrum, University of Basel, 4058 Basel, Switzerland. [5] Forage Science Group, AgResearch, Christchurch 8140, New Zealand. [6] Faculty of Science, Leiden University, 2300 RA Leiden, The Netherlands. [7] Food & Bio-based Products Group, AgResearch, Palmerston North 4442, New Zealand. [8] Institute for Glycomics, Griffith University, Gold Coast Queensland 4222, Australia. [9] School of Biological Sciences, University of Auckland, Auckland 1142, New Zealand. Correspondence and requests for materials should be addressed to M.J.L. (email: m.landsberg@uq.edu.au)

ABC toxins are secreted endotoxins produced by bacterial pathogens of insects and animals[1]. Originally described as Tcs (for toxin complexes), and more recently described as $A_5BC$ toxins in reference to their stoichiometry[2], all three subunits (A, B and C) are required to assemble a complex with full toxicity. Together, B and C form a well-characterised cocoon-like structure[3,4] that encapsulates a highly potent, non-selective cytotoxin, which largely determines the toxicity of the complex as a whole[5,6]. The cytotoxin is an auto-proteolytically cleaved C-terminal domain of the C subunit[3,4,7] that varies in sequence both across species, and within bacterial species harbouring multiple ABC toxin-encoding loci. The A subunit binds the B and C components[7,8] and following a conformational change, opens a transmembrane pore within lipid bilayers[2,8]. The pore is thought to facilitate translocation of the cleaved cytotoxin out of the BC-cocoon in an unfolded or partly unfolded state[4] and into the cytosol of targeted cells, a mechanism that at the supramolecular level shares some similarity with other multi-component pore-forming toxins[9,10]. In addition to pore formation, the A subunit also determines the host cell specificity of the holotoxin[11,12].

The Gram-negative bacterium _Y. entomophaga_[13] is a soil dwelling pathogen of insects that secretes an ABC toxin (YenTc) exhibiting potent oral toxicity towards a range of coleopteran species[14]. YenTcA is the pore-forming and membrane-binding module of YenTc. It has an unusual architecture amongst ABC toxins, in that it is encoded by four genes, two of which represent an apparently split tcA-like gene (_yenA1_ and _yenA2_) and two of which encode functional endochitinases (_chi1_ and _chi2_)[15]. All four genes are sequentially arranged within the major pathogenicity island of _Y. entomophaga_, along with genes that encode the B subunit (yenB1) and two, alternatively associated C subunits (yenC1 and yenC2)[14]. YenTcA contains five copies of each of the four proteins YenA1, YenA2, Chi1 and Chi2 and is thus comprised of 20 polypeptides in total, with a molecular weight exceeding 2 MDa[7,14]. Hereafter, we refer to the multi-A-chain YenTc-like toxins as type II ABC toxins, distinguishing them from simpler toxins, which have an A subunit encoded by a single gene (type I ABC toxins).

Previously we determined a low resolution structure of YenTcA[7]. The current manuscript reports a substantially improved structure of YenTcA isolated from a transposon mutant of the native _Y. entomophaga_, in its soluble (pre-pore) conformation, determined by single-particle cryo-EM and 3D reconstruction. The resolution of the structure, estimated at 4.4 Å, allows unequivocal assignment of all four unique polypeptide chains within the structure, with an overall atomic model able to be built that accounts for 75% of the complete amino acid sequence. We additionally determined a second structure using single-particle cryo-EM, which represents YenTcA stabilised in its pore form by insertion into POPC liposomes. This map has a reported average resolution of 11 Å, sufficient to guide the pseudo-atomic modelling of domain rearrangements that occur upon insertion of YenTcA into a lipid bilayer.

The structures of YenTcA provide insights into how the multiple subunits of type II ABC toxins are arranged to form a channel-forming translocation pore that is surprisingly similar in overall fold to the previously characterised type I ABC toxin, TcdA1[4], despite having a relatively low shared amino acid identity. Most significantly, key residues expected to regulate gating of the central translocation channel are not conserved between TcdA1 and YenTcA, pointing to possible differences in gating mechanisms. Moreover, both the pre-pore and pore structures of YenTcA identify Chi1 and Chi2 subunits—incorporation of which is a unique structural feature of YenTcA amongst characterised ABC toxins—as potential mediators of cell surface recognition. To this end, we present glycan binding analyses, the results of which indicate that YenTc, potentially via the Chi1 and Chi2 subunits, is able to bind a range of carbohydrate structures. To our knowledge, a cell surface receptor has yet to be identified for any ABC toxin and these data therefore represent evidence that ABC toxins harbour lectin activity and may therefore recognise cell surface glycans in vivo.

## Results

**Cryo-EM and structure determination of YenTcA pre-pore**. For all experiments described here, YenTcA was purified from the K9 strain of the native source bacterium _Y. entomophaga_[7]. Working with the natively expressed YenTcA from _Y. entomophaga_ not only overcomes problems associated with producing this toxin recombinantly, but also provides reassurance that the structure is not artificially perturbed by fusion tags or production of the complex in a non-native host. Using cryo-EM in combination with single-particle image processing, we were able to determine a structure for YenTcA at an average resolution of 4.4 Å (Fig. 1a–d, Table 1, Supplementary Fig. 1). As is commonly observed with cryo-EM maps, the resolution varies throughout the map, meaning substantial parts of the structure could be built de novo including most of the YenA1 and YenA2 chains, for which no previous structural information was available (Supplementary Fig. 2). Densities corresponding to Chi1 and Chi2 were generally less well resolved, but models of these regions were obtained by initially fitting crystal structures of the chitinase domains (PDB IDs 3oa5, 4dws)[3,15] into the map as rigid bodies and, in the case of Chi2, subsequently refining the crystal structure coordinates together with the rest of the model using interactive molecular dynamics. The final atomic model (Fig. 1e, f) represents 75% of the combined amino acid sequences (Table 1).

YenTcA is assembled from five symmetrically-arranged protomers, each comprising a single copy of YenA1, YenA2, Chi2 (Fig. 1f) and Chi1. The overall structure can be roughly partitioned into four distinct regions (Fig. 1). Each protomer contributes a pair of extended α-helices that form a funnel-shaped longitudinal pore at the core of the complex that widens into a β-sandwich domain (the TcB-binding domain) at one end and is enclosed at the opposite end by a ring of five neuraminidase-like domains (again, one is contributed by each protomer). This structure—the pore-forming apparatus—is entirely encoded by the YenA2 chain. A predominantly α-helical shell incorporating regions of both YenA1 and YenA2 forms the next layer, encapsulating the longitudinal pore. The Chi2 proteins form an equatorial ring around the lower half of the α-helical shell, while five, leg-like protrusions emerge from the base of the complex. The densities corresponding to this latter feature are relatively diffuse in the 2D class averages (Supplementary Fig. 1b) suggesting these correspond to structural elements that do not occupy a single, fixed location.

YenA2 contributes a total of 21 resolved α-helices to the outer shell (residues 1–165; 559–812) while YenA1 contributes a total of 45 resolved α-helices (residues 45–304; 584–1164). There are substantial interactions between YenA1 and YenA2, both within a single protomer and between neighbouring protomers (Supplementary Fig. 3). The molecular interface between YenA1 and YenA2 within a single protomer represents a buried surface area of 2709 Å² (Supplementary Fig. 4a,b) and the C-terminus of YenA1 is in close proximity to the N-terminus of YenA2 (separated by approximately 16 Å; Supplementary Fig. 4c). When the Chi2 and Chi1 proteins are excluded from the model, some structural similarity is apparent between the combined YenA1 and YenA2 chains, and the type I TcA protein, TcdA1 from _P. luminescens_, for which a 4.0 Å X-ray crystal structure has been

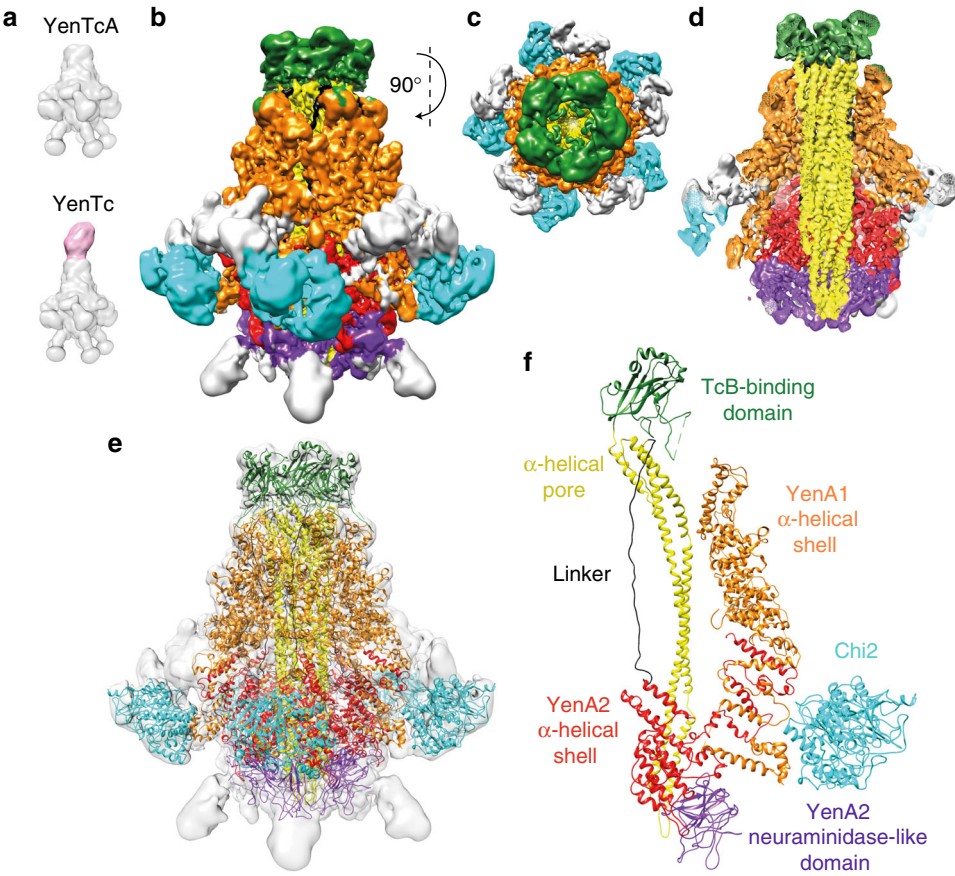

**Fig. 1** The structure of YenTcA at 4.4 Å resolution. **a** Schematic representing the structure of YenTcA in the context of the holotoxin, YenTc. **b**, **c** 3D Cryo-EM map of the pentameric YenTcA in the soluble (pre-pore) conformation as side and top view orientations. **d** A cut-away view of the map revealing the encapsulated translocation pore. The map (**b**–**d**) is filtered according to local resolution and coloured according to the structural subdomains subsequently identified in the atomic model, which is displayed with the map transparently overlayed in **e**. A single protomer is shown in **f**. Colouring: YenA1 α-helical shell domain (orange); YenA2 TcB-binding domain (green), α-helical pore domain (yellow), linker (black), α-helical shell domain (red), neuraminidase-like domain (purple); Chi2 chitinase domain (cyan). Density corresponding to Chi1 is masked out in this map (refer to Fig. 2). Areas coloured grey correspond to regions of the map where it was not possible to build an atomic model due to structural ambiguity and/or the low local resolution of the map in these regions

determined previously[4] (Supplementary Fig. 4d). The degree of similarity was somewhat unexpected given the relatively low amino acid sequence identity: YenA1 (130 kDa) and the N-terminal half of TcdA1 are 17.6% identical while YenA2 (156 kDa) and the C-terminal half of TcdA1 are more similar, sharing a 26.8% identity. The pore-forming apparatus clearly has a very similar overall fold between the two toxins, explaining the higher level of sequence identity between the YenA2 chain and the corresponding region of TcdA1. Given the close association of the YenA1 and YenA2 chains, the proximity of their respective C- and N- termini and the unexpectedly high degree of overall structural similarity to the type I TcdA1 protein, we concluded that the two subtypes of ABC toxins have arisen as a result of an evolutionary gene fusion or gene fission event.

**Location of the endochitinases, Chi1 and Chi2.** Chi1 and Chi2 are encoded by genes that flank *yenA1* and *yenA2* within the pathogenicity island. Our previously determined crystal structure of the chitinase domain of Chi2 (residues 92–632)[3] was unambiguously assignable into the five symmetrically-arranged densities located at the equatorial periphery of the cryo-EM map (cyan densities in Fig. 1). A region of density for which no model could be built (Fig. 2a) represents the primary interface between Chi2 and YenA1. However, based on the orientation of the Chi2

chitinase domain, this interface is predicted to involve the N-terminal domain of Chi2 (residues 1–91) and a domain insertion between helices A1α14 and A1α15 of the α-helical shell, corresponding to residues 305–583 of YenA1 (Supplementary Fig. 3). More explicit definition of the interface was not possible since neither region shares detectable homology with existing high resolution structures, and the local resolution of the cryo-EM map was insufficient to allow unambiguous model building de novo. Residues 46–57 (α1), located towards the N-terminus of the neighbouring YenA1 protomer form a second interface. Chi2 is incorporated into the overall structure such that its substrate cleft is solvent exposed and opens approximately downwards. A conserved DxDxE motif ($^{345}$DIDWE$^{349}$ in Chi2) that sits deep within this substrate cleft represents the enzyme's active site (coloured magenta in Fig. 2b).

Initially we were unable to locate density corresponding to Chi1 in our high resolution cryo-EM map, but inspection of maps generated in early stages of the refinement revealed a further five densities extending from the base of the complex, attached like feet to the five, apparently mobile leg densities that extend beneath the main body of the complex (Fig. 2c). These features were masked out in the high resolution map used for model building. The map density in this region suggests a much lower local resolution than the rest of the complex, explaining their absence from the final map shown in Fig. 1. The corresponding

**Table 1 Structure determination and model building statistics**

|  | YenTcA pre-pore form (EMDB-20053) (PDB 6ogd) | YenTcA pore form (EMDB-20054) |
|---|---|---|
| *Data collection and processing* |  |  |
| Magnification | 22,500× | 23,000× |
| Voltage (kV) | 300 | 300 |
| Electron exposure (e- Å$^{-2}$) | 80 | 32–37 |
| Defocus range (μm) | −0.5 to −2.5 | −2.0 to −4.0 |
| Pixel size (Å) | 1.34 | 1.72 |
| Symmetry imposed | C5 | C5 |
| Initial particle images (no.) | 19,713 | 68,000 |
| Final particle images (no.) | 9856 | 5130 |
| Map resolution (Å) | 4.4 | 11 |
| FSC threshold | 0.143 | 0.143 |
| Map resolution range (Å) | 3.9–10.4 | n/a |
| *Refinement (excluding Chi1)* |  |  |
| Initial model used (PDB code) | n/a | n/a |
| Model resolution (Å) | 6.4 (vs whole map) |  |
| FSC threshold | 0.5 |  |
| Map sharpening *B* factor (Å$^2$) | n/a | n/a |
| Model composition |  |  |
| Non-hydrogen atoms | 94,125 |  |
| Protein residues | 11,820 |  |
| Ligands | 0 |  |
| R.m.s. deviations |  |  |
| Bond lengths (Å) | 0.014 |  |
| Bond angles (°) | 1.986 |  |
| Validation |  |  |
| MolProbity score | 1.68 (hundredth percentile) |  |
| Clashscore | 1.38 (hundredth percentile) |  |
| Poor rotamers (%) | 1.99 |  |
| Ramachandran plot |  |  |
| Favored (%) | 88.48 |  |
| Allowed (%) | 9.86 |  |
| Disallowed (%) | 1.66 |  |

density is clearly seen in reference-free class averages (Fig. 2d, pink arrows, also see below), confirming that these poorly-resolved densities represent real structural features and not, for example, a masking artefact or a consequence of over-refined noise. While the orientation of Chi1 within these densities could not be unambiguously determined, rigid fitting (Fig. 2c) indicates that their size is consistent with our previously determined crystal structure of Chi1[15].

**Properties of the YenTcA pore-forming apparatus**. The central, funnel-shaped density formed by the five extended α-helical pairs of YenA2 (residues 862–1170) represents a pre-formed translocation pore (Fig. 3a) that connects to the B and C subunits in the wild-type holotoxin via the TcB-binding domain (residues 1171–1343)[7]. The ~250 Å long pore varies between 10 and 30 Å diameter over the first 150 Å before narrowing to essentially complete closure at the tip (Fig. 3b, c). As mentioned already, the overall fold is similar to the pore-forming apparatus of TcdA1, but there are some significant differences in amino acid residues lining the pore. Most notably, a ring of five histidines (H996) forms a constriction (to ~6 Å diameter) about 20 Å above the tip, while the final closure involves hydrophobic interactions of the sidechains of the five L990 residues (Fig. 3c, d). The histidine sidechains face inwards and are particularly well resolved in

the map, suggesting a rigid arrangement. Based on the observed spacing and geometry inferred from the map density, we were able to perform molecular modelling and establish that a ring of stabilising ($N_\delta$–OH–$N_\varepsilon$) hydrogen bonds is formed involving a single bridging water molecule between each sidechain (Supplementary Fig. 5). Such an arrangement is dependent on all five sidechains remaining deprotonated, suggesting that reduction of the pH below the effective pKa of the histidine sidechain (typically ~pH 6) may play a role in triggering expansion of the pore. While this hypothesis could not be validated experimentally (e.g. by site-directed mutagenesis, due to the lack of a recombinant expression system for YenTcA), the suggestion is entirely consistent with a pH-driven pore formation mechanism, as has been proposed previously for ABC toxins[2,8], and also seen in other bacterial pore-forming toxins[16–18]. The release of the cytotoxic C subunit from the TcB-like protein in the context of the YenTc holotoxin is also thought to be pH regulated[3].

Histidine residues are known to be key drivers of pH-dependent conformational change[19] as well as channel gating[20,21] in functionally diverse proteins. What is notable in this context however, is that H996 in YenA2 is not absolutely conserved in other ABC toxin homologues. We examined the sequences of a number of other TcA proteins and could find only two where the corresponding residue was a histidine (the TcdA2 proteins from *P. luminescens* and *P. asymbiotica*) (Supplementary Fig. 5b,c). In all the other sequences examined, a strongly conserved arginine was found. Indeed, in the pre-pore structure of TcdA1 (PDB ID: 4o9y), the equivalent arginine forms an extensive salt bridging network involving a number of residues that are not conserved in YenA2. Comparison of these two structures thus reveals that the chemical environment at the pore closure is quite different between YenTcA and TcdA1, indicating that they are likely gated via different mechanisms.

The five neuraminidase-like domains, formed by two non-contiguous regions of the YenA2 polypeptide (166–332; 523–558) form a seal beneath the translocation pore. An arrangement of similar domains is also seen in the pre-pore structure of TcdA1[4]. Interactions between the neuraminidase-like domains in this region are mediated by an extended hairpin loop (Fig. 3e, YenA2 residues 244–278—the pore-closing loop) that reaches towards the five-fold axis of the complex. The equivalent loop in TcdA1 has been proposed to form an electrostatic lock that stabilises the pre-pore form of the complex[4]. Interestingly, the amino acid sequence comprising these loops is poorly conserved across TcA proteins (Fig. 3e) suggesting that the stability or susceptibility of the electrostatic lock to different chemical environments is likely to vary between TcA-like proteins. Together with the unconserved nature of the H996 substitution discussed above, this observation adds further weight to the hypothesis that release and opening of the YenTcA pore may be triggered by unique physiological conditions, or at the very least involves a mechanism that is different to that which has currently been proposed for TcdA1.

**Structural changes accompany YenTcA insertion into liposomes**. To further investigate possible drivers of pore formation, we performed electrostatic surface potential calculations in silico over a range of pH values (Supplementary Fig. 6). These calculations predict that repulsive forces in the pore-closing loop region dominate over a wide pH range, including under the conditions at which our pre-pore structure was determined (pH 7.5). While there is some ambiguity associated with these calculations, in particular due to the presence of several long, potentially flexible amino acid sidechains within the pore-closing loop, this nonetheless led us to consider the hypothesis that

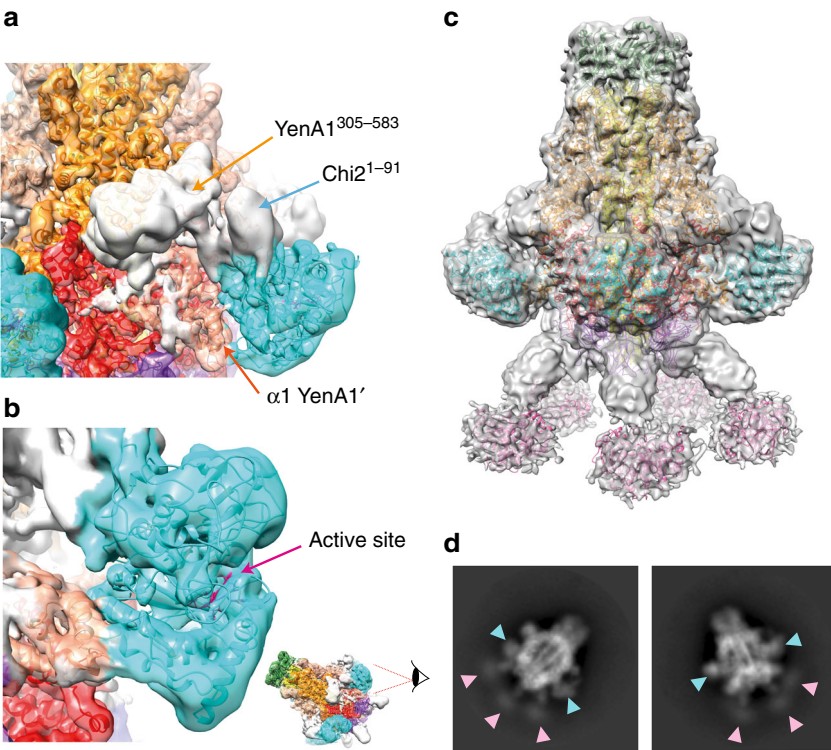

**Fig. 2** Chitinases in the context of the pre-pore structure. **a** Two unmodelled subdomains form the major interacting surface between YenA1 and Chi2: an insertion into the YenA1 α-helical shell (YenA1$^{305-583}$) and the N-terminal domain of Chi2 (Chi2$^{1-91}$). A secondary interaction surface involves the α-helical shell (α1) from the neighbouring protomer (YenA1′). **b** The Chi2 active site (magenta) sits at the bottom of a downward facing substrate cleft. The Chi1 subunit could not be accommodated within any of the unbuilt regions of the final, high resolution map (Fig. 1) but densities that corresponded in size to the Chi1 crystal structure (PDB ID 3oa5) were resolved in maps produced during earlier refinement iterations (**c**; X-ray structures coloured in pink and fitted as a rigid body) and were also clearly evident in reference-free class averages (**d**; pink arrowheads)

altering the surface potential of the electrostatic lock alone may not be sufficient to regulate the pre-pore to pore transition in YenTcA. Indeed, repeated attempts to obtain an alternative conformation of YenTcA by pH titration in vitro have been unsuccessful in our hands. Using electrophysiological techniques (Supplementary Fig. 7) similar to those employed previously to demonstrate the pore-forming properties of the *P. luminescens* TcdA1 toxin[22], we were however able to demonstrate that both YenTcA and the YenTc holotoxin form ion-permeable transmembrane pores. Following on from this, we were able to stabilise a membrane-inserted conformation of YenTcA in vitro by incubation with POPC liposomes (Supplementary Fig. 8).

Using cryo-EM and single-particle analysis, we obtained a 3D reconstruction of the membrane-inserted YenTcA at 11 Å resolution (Fig. 4a, b, Supplementary Fig. 8). While this resolution is not sufficient to allow atomic model building de novo, we were able to unequivocally fit several subunits and structural subdomains into the map as rigid bodies, including the pore domain, the Chi1 and Chi2 chitinase domains, the neuraminidase-like domains as well as a region of the YenA1 α-helical shell located furthest from the membrane. Rearrangements involving key structural domains are clearly evident from these models (Fig. 4c, d) and were even interpretable in 2D classes (Fig. 4e, f), particularly when these were compared to the equivalent data for the soluble pre-pore structure (Fig. 4g, h).

Exposure of the previously encapsulated translocation pore is the most immediately striking aspect of the conformational change. In the pre-pore structure, the strained YenA2 linker (residues 813–861) connects the top of the translocation pore to the bottom of the α-helical shell. Like TcdA1[2], this linker also lacks any regular secondary structure in the pre-pore state, and is

similar in length (49 residues versus 47 residues for TcdA1), distance covered (~115 Å versus ~110 Å) and sequence (~55% conserved) to the TcdA1 linker, so it would appear most likely that entropically-favoured linker collapse is the most likely driver of membrane insertion in YenTcA.

The other notable structural rearrangement is unique to YenTcA. It involves an outward rotation of the Chi1 subunit densities, a movement that is necessitated by the fact that in the pre-pore configuration (Fig. 2c, d; Fig. 4g, h) the Chi1 subunits are positioned directly beneath what becomes the membrane proximal surface of YenTcA. In this location, Chi1 would sterically hinder protrusion of the pore from the surrounding shell of the complex. In the membrane-inserted structure, the Chi1 densities (clearly visible in both class averages (Fig. 4e, f) and in the 3D reconstruction (labelled in Fig. 4a–d)) have moved parallel to the putative membrane surface by ~100 Å (Supplementary Movie 1). This rearrangement provides space for the protruding pore, allowing it to insert into a lipid membrane, while maintaining a close proximity of the Chi1 subunits to the membrane surface.

**Implications for receptor recognition and host tropism**. The Chi1 subunit of YenTcA is of further interest as it is indirectly coupled to the pore-closing loop of the neuraminidase-like domain; it interacts with YenA2 via a region that maps to residues 333–522, corresponding to an insertion also within the neuraminidase-like domain (labelled as the knee loop in Supplementary Fig. 9). Comparing YenTcA to the previously characterised TcdA1 reveals that a slightly longer sequence (~270 residues) is also inserted into the corresponding neuraminidase-like domain, and folds

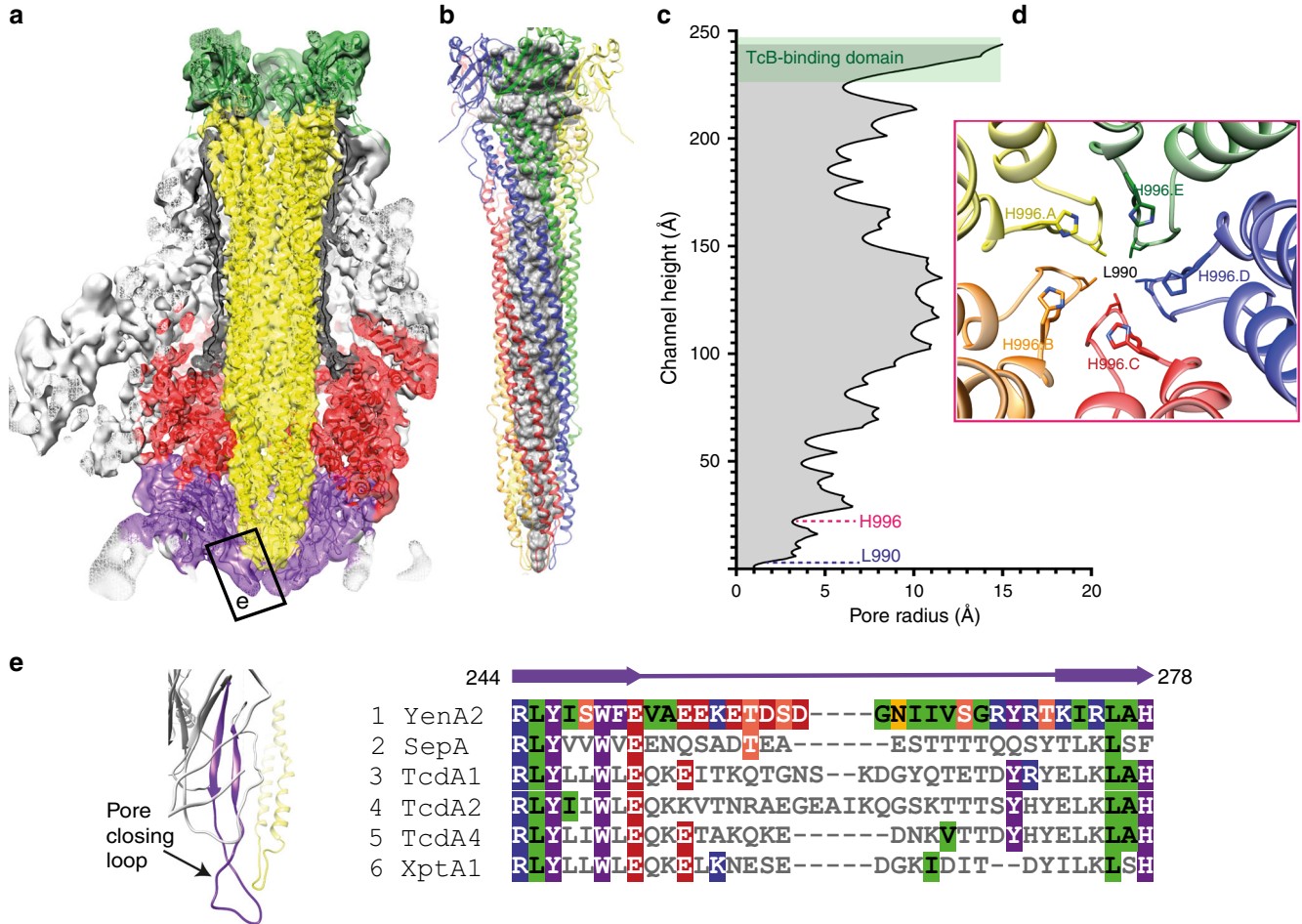

**Fig. 3** Properties of the encapsulated YenTcA translocation pore. **a** A central section of the YenTcA pre-pore structure highlighting the pentameric translocation pore (yellow) as well as the neuraminidase-like domains that close the bottom of the complex (purple). **b, c** The translocation pore has a maximum radius of 15 Å and narrows towards the bottom, constricted by H996 (also shown as viewed from above in **d**) and L990. In **b, d**, the helices are coloured by protomer. **e** The extended hairpin loop structure of the neuraminidase-like domain that seals the pore (highlighted in purple). The loop sequence in particular is not well conserved with the equivalent sequences of TcA proteins from other ABC toxins

into two IgG-like domains which were identified by the authors of a previous study as putative receptor-binding domains[4]. Despite both Chi1 (in the case of YenTcA) and the IgG-like domains (in the case of TcdA1) being coupled to the respective pore-closing loops, their positions relative to the rest of the structure are somewhat different (Supplementary Fig. 10). Unlike the IgG-like domains in the TcdA1 structure, Chi1 extends downwards, away from the structural shell but encircling the region from which the translocation pore ultimately protrudes, a location that seems even better suited to facilitating initial recognition of, and binding to, a cellular membrane. It is therefore tempting to suggest that Chi1 may play an important role in cellular recognition by YenTc. The fact that Chi1 shares no structural similarity with the putative receptor-binding domains of TcdA1 also establishes that ABC toxins generally are capable of incorporating markedly different structural domains, which seems a likely mechanism for establishing host tropism.

An increasing body of evidence suggests that cell surface glycans play important roles in cellular recognition by bacterial toxins[23–26] and we noted that the structure of YenTcA contained a number of candidate lectin domains—including Chi1 and Chi2 as well as the neuraminidase-like domain. We therefore performed a high-throughput glycan microarray screen[27]. We examined the glycan binding profiles of YenTc, as well as the Chi1 and Chi2 subunits in isolation (Fig. 5, Supplementary

Fig. 11). Out of 423 unique glycan structures, we observed significant binding of all three proteins/complexes tested to a range of structures including those incorporating galactose (Gal), glucose (Glc), N-acetylgalactosamine (GalNAc) and N-acetylglucosamine (GlcNAc) motifs. β1–4 linked homopolymers of GlcNAc are the natural substrate for chitinases, and we previously established that both Chi2 and, to a lesser extent, Chi1 hydrolyse internal β1–4 linkages of such polymers[15] but we noted that the pH optimum for both enzymes identified in this study was shifted away from the alkaline pH environment of the insect midgut[28]. Thus, it cannot be ruled out that in vivo, Chi1 and/or Chi2 may bind to (but not hydrolyse) GlcNAc-containing structures. Both GlcNAc and GalNAc are major components of glycophospholipids in insects[29–31] and so binding to these in vivo may occur in the context of binding to glycolipids that form part of the host gut membrane.

Binding to fucosylated structures and mannosyl derivatives was also observed, but was much more prevalent in the glycan binding profile of YenTc than either of the chitinase subunits alone. Mannose is a component of the basal cell membrane in insects and is also incorporated into larger O- and N-linked glycan structures found in insects[31–33]. Fucosylated glycans are also prevalent within the glycobiome of insects[31,34] and the prevalence of fucosylated structures within the glycan binding profile of YenTc is therefore completely consistent with YenTc

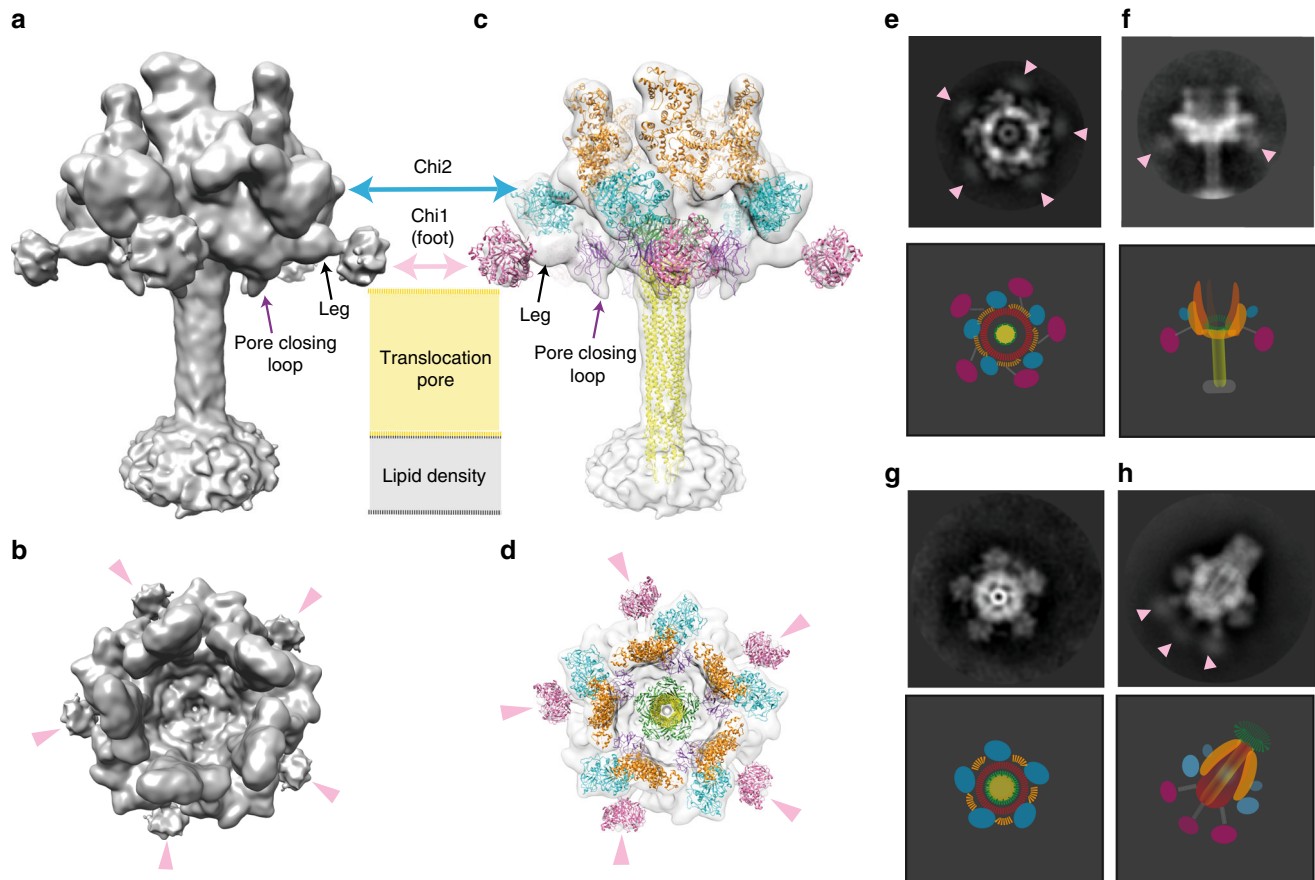

**Fig. 4** Structural analysis of YenTcA inserted into liposomes. **a, b** 3D reconstruction of YenTcA incorporated into POPC liposomes at an estimated resolution of 11 Å. **c, d** Pseudo-atomic model of the YenTcA pore form using the same map shown in **a, b**: despite low resolution, domains of the YenTcA pre-pore atomic model (Fig. 1) could be rigidly fitted into representative densities of the YenTcA pore form map: Chi1 (pink), Chi2 (cyan), pore (yellow), TcB-binding domain (green), neuraminidase-like domains with pore-closing loop (purple) and a substantial part of the A1 shell domain (A1$_{236–1161}$, orange). The map, as well as comparative 2D class averages of the pore (**e, f**) and pre-pore forms (**g, h**) reveals clear structural rearrangements evident in the liposome-inserted map, most strikingly the protrusion of the translocation pore from the base of the complex and the outward rotation of the Chi1 subunits (pink arrowheads in **b, d, e** and **f**). The class averages are representative top (**e**) and side views (**f**) of the pore form, and top (**g**) and side views (**h**) of the pre-pore form. Beneath each of the class averages is a schematic representation of the complex that indicates the location of different structural subdomains. Colouring mirrors the colour scheme throughout the manuscript: orange—YenA1 α-helical shell; red—YenA2 α-helical shell; green—TcB-binding domain; yellow—pore domain; cyan—Chi2; pink—Chi1

being responsible for disease in insect hosts. There were six hits in common between the glycan binding profiles of Chi2 and YenTc (Fig. 5b, c) of which two represent branched, fucosylated glycan structures (Fig. 5c, Supplementary Data 1; compound IDs 538 and 542). A fucosylated pentasaccharide was also one of the common hits between Chi1 and YenTc. Exploring the potential of these compounds to act as receptors (or components of a receptor) for cellular targeting of YenTc remains a goal, but has been hindered to date by the inability to produce YenTc (or mutants thereof) in recombinant expression systems, as well as the limited tools for performing genetic manipulations directly in *Y. entomophaga*.

Consistent with YenTc being an insect-specific toxin, all three profiles showed limited examples of binding to glycan structures that are specific to vertebrates. Binding was observed to only a relatively small number of the sialylated structures included in this screen (all of which contained N-acetylneuraminic acid, the predominant sialic acid in humans) and none of the sialylated structures bound by Chi1 or Chi2 were reproduced in the YenTc glycan binding profile. The most probable explanation for the observed hits is that these instead represent low affinity binding to other chemical groups (for example, all of the hits recorded against sialylated structures featured either a Gal, GlcNAc or Fuc group). Similarly, very few examples of binding to complex (vertebrate) N-linked glycans were observed.

## Discussion

The identification and structural characterisation of ABC toxins as virulence factors in pathogenic bacteria is a relatively recent development. The TcA subunit is critical for cell surface recognition and the introduction of a toxin-translocating transmembrane pore. To date, everything that has been learned about the mechanistic details of this process has been deduced from a single characterised example—the TcdA1 protein, one of several type I TcA-like proteins found in *P. luminescens*[2,4,8]. Here we have described, at near atomic resolution, the structure of YenTcA—the type II A subunit from the orally active ABC toxin, YenTc. A hallmark of the type II ABC toxins is the split A gene architecture, with the resulting protein structure characterised here in the structure of YenTcA. We observed similarity in the overall fold of TcdA1 and YenTcA, which was unexpected based on the level of sequence conservation, although it is worth noting that the highest degree of sequence similarity was seen between the YenA2

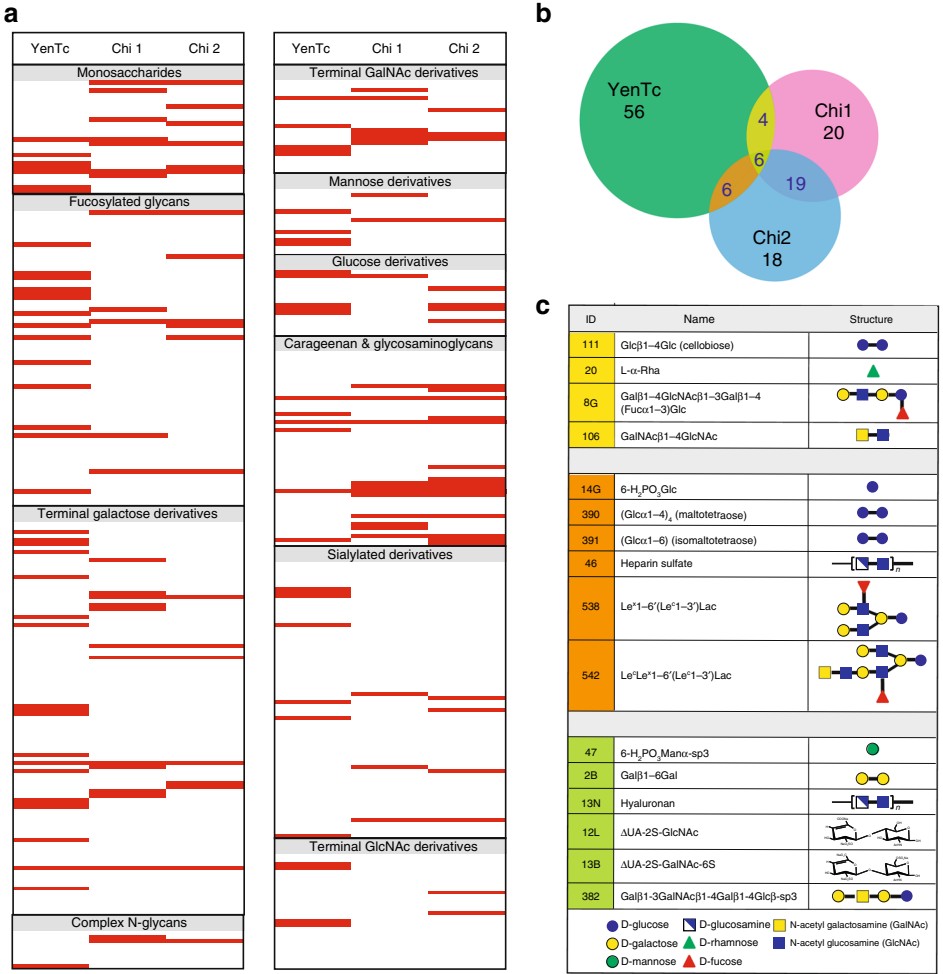

**Fig. 5** Evidence for lectin-like properties of YenTc. **a** Binding profile of the glycans grouped according to glycan property. Each red line indicates a hit to a glycan. The raw data used to produce the heat map shown in panel **a** are presented in Supplementary Fig. 11 and the full list of glycans printed on the arrays is given in Supplementary Data 1. **b** Venn diagram representation of overlapping hits between all three screens. In total, 72 glycan structures bound to YenTc including 56 that were unique to YenTc (in green). Chi1 and Chi2 both bound 49 glycan structures in total with 20 structures uniquely recognised by Chi1 (in pink) and 18 uniquely recognised by Chi2 (in blue). Four structures were recognised by both Chi1 and YenTc (yellow), the structures of which are represented in **c** and six structures were recognised by both Chi2 and YenTc (orange), also represented in **c**. Structures that were recognised by all three proteins are also shown in panels **b** and **c** (olive)

chain (which encodes the pore-forming domain) and the corresponding, C-terminal half of the type I TcA-like protein, TcdA1. An open question remains whether these two structural subtypes of ABC toxins have arisen as a result of evolutionary gene fusion or gene fission although large-scale genomic analyses have found that fusion is four times more common than evolutionary gene fission[35], providing some indication that type II ABC toxins may represent the more ancient arrangement.

Our studies have provided a framework for understanding the structural mechanisms through which ABC toxins may evolve new specificities. The pore-forming domains of both TcdA1 (reported previously) and YenTcA (described here), which share some structural conservation, are decorated by diverse structures implicated through their fold and location in regulating interactions with host receptors. This in turn gives rise to diverse patterns of host or cell tropism. Sequences inserted into the α14-α15 loop within YenA1, and into the knee loop of the neuraminidase-like domain of YenA2 are directly implicated in binding of the Chi2 and Chi1 subunits that are a unique structural feature of YenTcA. These domain insertion points are completely conserved with the insertion points of the putative receptor-binding

domains within the structure of *P. luminescens* TcdA1. It therefore appears that these insertion points are particularly susceptible to the evolutionary transfer of sequences that either directly, or via recruitment of additional subunits, define the tropism of ABC toxins. We also identified broad structural rearrangements that occur following insertion of YenTcA into a lipid bilayer, including the repositioning of Chi1 in a way that maintains close proximity to the membrane but allows protrusion of the previously ensheathed translocation pore. Collectively, the evidence we present here is entirely consistent with the hypothesis that variable domains either directly or indirectly facilitate receptor recognition. Thus our findings contribute to an emerging paradigm in pore-forming toxin biology generally that appears to also hold true for ABC toxins specifically—that classes of pore-forming toxins are defined by sequence and structural similarity that is usually confined to the pore-forming apparatus, with specificity established by variable, auxiliary domains.

YenTc targets midgut epithelial cells in susceptible insects[6,7] and is a major determinant of the oral toxicity associated with *Y. entomophaga*. The semi-permeable peritrophic membrane of the insect midgut is rich in chitin and protects the epithelial cell

layer. Co-secretion of chitinases is a strategy commonly employed by microbial pathogens to overcome the barrier imposed by the peritrophic matrix[36] but we have presented evidence here that the chitinase subunits may fulfill an alternative functional role. It is plausible that the chitinase activity of YenTc in vivo may be regulated by pH gradients. Furthermore the chitinase activity of YenTc in vitro appears to be completely accounted for by the in vitro activity of five Chi2 subunits[7,15]. Thus Chi1, which we identified as being located closest to the membrane surface following membrane bilayer insertion, may not be required to fulfil a catalytic role and may instead be primarily involved in cell surface recognition. Binding to cell surface glycans may also involve a combinatorial recognition mechanism, involving other lectin domains, which might explain the observed binding of YenTc to some classes of glycans that are not reproduced in the glycan binding profiles of the chitinase subunits alone. The importance of glycan-protein interactions in cellular recognition processes is an emerging paradigm[37] and it has already been established that glycan receptors are targets for many pore-forming toxins, including several of the so-called binary toxins that are analogous to the YenTc system in that they combine a pore-forming domain, receptor-binding domain and an enzymatic domain[24–26,38,39]. Our studies have provided evidence that glycans may also play an important role in cellular recognition by ABC toxins.

In summary, our structural and functional data inform a refined model for the mechanism of host recognition and conformational change associated with the YenTc holotoxin in vivo, which may extend to other ABC toxins in general (Fig. 6). YenTc is thought to initially recognise one (or more) cell surface receptors within susceptible hosts that may be glycans in nature. Based on our studies described here, it is now established that the pore-forming apparatus is a structural scaffold, conserved amongst ABC toxins,

which is capable of accommodating diverse structural motifs and thus appears highly amenable to evolutionary adaptation in response to host or environmental changes. This is an important consideration, given that genes encoding ABC toxins have been identified in a wide range of bacterial pathogens, including those that target vertebrate rather than invertebrate hosts[40,41]. How pore formation is triggered in vivo remains to be unequivocally established, despite some evidence that pH changes may play an important role[4]. We propose that toxin-receptor interactions may induce structural changes that are transferred (e.g. allosterically) to the pore-closing loops, contributing to conformational change and membrane insertion. Putative receptor-binding domains with possible lectin activity remain in closest proximity to the host membrane following conformational change, supporting a role for these domains in binding to cell surface receptors that contain glycan structures. Future identification of a cellular receptor for YenTc will undoubtedly lead to an enhanced understanding of its associated insecticidal activity, also offering the potential to guide strategies for manipulating host specificity for biotechnological and biomedical applications.

## Methods

**Bacterial strains and protein purification**. The *Yersinia entomophaga* K9 strain used in this study is maintained in the Hurst laboratory (AgResearch NZ). The parent strain, *Y. entomophaga* (MH96), was isolated from the diseased larvae of a native New Zealand grass grub (*C. zealandica*). The K9 stain contains a transposon insertion that prevents the correct transcription and translation of the YenB gene and thus secretes a toxin complex that only contains the YenTcA components (YenA1, YenA2, Chi1 and Chi2)[14]. The procedures for isolation and purification of YenTcA for structural studies followed those we have developed previously[7,42]. *Y. entomophaga* K9 cells were grown in LB medium supplemented with kanamycin (50 μg ml$^{-1}$) overnight at 25 °C. Following centrifugation, the media supernatant was subjected to ammonium sulfate precipitation using a final concentration of ~70% w/v, and the precipitate was resuspended in Tris buffer (25 mM Tris, 150 mM NaCl, pH 7.5). The resuspended protein pellet was further purified by

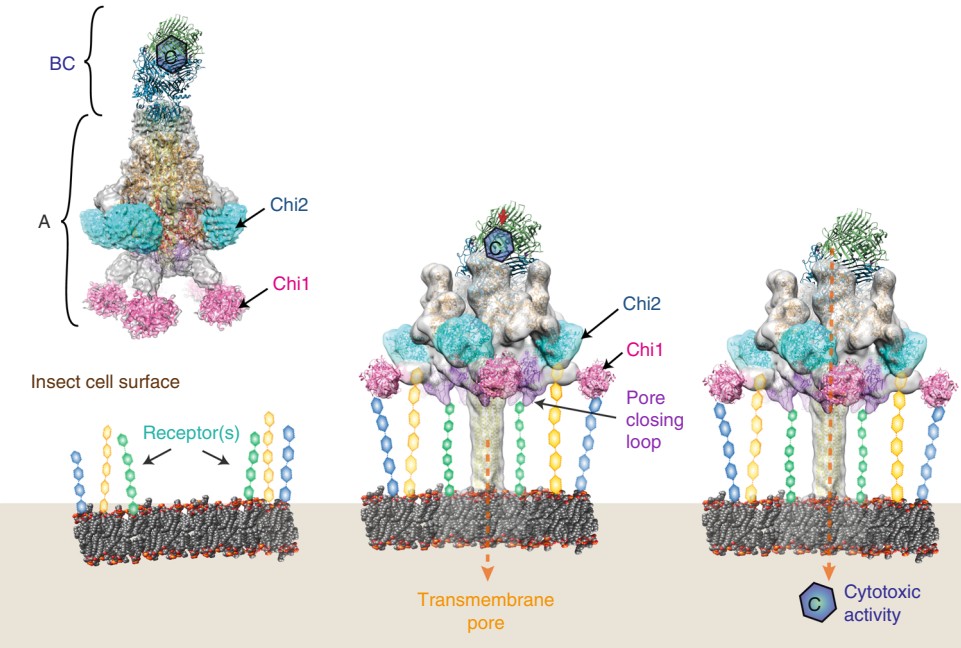

**Fig. 6** Proposed mechanism of YenTc pore formation and toxin translocation. The YenTc recognises receptor(s) on the surface of insect midgut epithelia. In the illustration, the receptors are represented as lipid-linked glycans but the precise nature of these remains to be identified. It is also possible that the specificity of YenTc may require binding to some combination of lipid-linked, protein-linked and/or protein receptors. A conformational change occurs— either prior to, upon or subsequent to receptor recognition—exposing the translocation pore, which is then inserted into the lipid bilayer. It is currently assumed that pore formation occurs within vesicles following endocytosis. The BC cage releases the toxin payload (encoded by the C-terminal sequence of the C subunit), which is delivered to susceptible cells through the translocation pore (note that the location of the BC subunit crystal structure is not based on structural modelling and is added solely for visualisation purposes)

size-exclusion chromatography. The fraction containing YenTcA protein was collected and subsequently concentrated for electron microscopy studies. Prior to the collection of larger datasets for high resolution structure determination, the quality of purified YenTcA protein was assessed using negative-stain electron microscopy.

**Cryo-EM data collection and image processing**. For the YenTcA pre-pore structure, cryo-EM grids were prepared using a Mark IV Vitrobot (ThermoFisher Scientific Inc.). Four microlitres of YenTcA (0.1 mg mL$^{-1}$ in 25 mM Tris, 150 mM NaCl, pH 7.5) to which NDSB-195 (final concentration 10 mM) had been added, was applied to an ethyl-acetate washed and glow-discharged Quantifoil® holey carbon grid (R2/2), allowed to absorb for 30 s and then blotted for 2 s before plunge freezing.

Data collection was performed on a Titan Krios (300 kV, $C_s$ 2.7 mm) (ThermoFisher Scientific Inc.) fitted with a K2 Summit direct detection device (Gatan Inc). Micrographs were taken at a nominal magnification of ×22,500, corresponding to a pixel size of 1.34 Å at the specimen level. Images were acquired in electron counting mode (4k mode) applying dose fractionation with a total of 40 sub-frames collected, representing a total dose of 80 e$^-$ Å$^{-2}$ (dose rate 7 e$^-$ px$^{-1}$ s$^{-1}$). The defocus ranged from −0.5 to −2.5 μm.

Correction for beam-induced motion was initially performed in real time as described in Scherer et al[43]. Fifteen thousand particles were picked using the EMAN2 Swarm particle selection algorithm[44]. The coordinates were imported into Relion[45] and auto refinement with imposed C5 symmetry (following 3D classification with 8500 particles taken forward) yielded a preliminary reconstruction with an estimated resolution of 8.2 Å (at the criterion FSC = 0.143). The unmasked map at this stage of the processing is shown in Fig. 2c. Further refinement of this data led to masking out of the feet densities and yielded a map with an estimated resolution of 5 Å inside the mask, representing a total of 6500 particles.

To improve this map, motion correction was performed together with dose-weighted averaging on the uncorrected image stacks using the Unblur program[46]. At this time, all subsequent image processing was carried out in EMAN2 (version 2.2)[47]. Two hundred two micrographs were selected after initial screening of images. Nineteen thousand seven hundred thirteen particles were picked using e2boxer and extracted with a square box size of 416 pixels. For CTF estimation, e2ctfauto was used. Reference-free class averaging was performed on Fourier cropped particle images that had been low-pass filtered to 15 Å. Particles falling into classes that clearly were not consistent with YenTcA (e.g. crystalline ice contamination) were excluded, leaving 18,931 particles for the initial gold standard structure refinement, a number subsequently reduced to 9856 for the final, high resolution refinement using the e2evalparticles tool. All refinements were carried out using implicit C5 symmetry. Resolution was estimated by Fourier shell correlation at the FSC = 0.143 criterion.

For the YenTcA pore structure, YenTcA at 1.2 mg/ml was mixed with POPC liposomes at a ratio of 1:2 (w/w). The protein-liposome mix was subjected to 3–5 freeze/thaw cycles for 2 min each (at liquid nitrogen temperature and 50 °C, respectively). Samples were centrifuged at 14,000 × $g$ for 10 min before plunge freezing. Cryo-EM grids were then prepared using a Mark II Vitrobot (FEI Company). Four microlitres of sample was applied to an ethyl-acetate washed, glow-discharged Quantifoil® holey carbon grid (R1.2/1.3), incubated for 3 s or 10 s before blotting for 3 s and plunge freezing in liquid ethane.

Data collection was carried out on a Tecnai F30 FEG-TEM (300 kV, $C_s$ 2.0 mm) (ThermoFisher Scientific Inc.) fitted with a K2 Summit direct detection device (Gatan Inc.). Micrographs were taken at a nominal magnification of ×23,000, corresponding to a pixel size of 1.72 Å at the specimen level. Data were collected in electron counting mode (4k mode) using dose fractionation, with a total of 40 subframes per image representing a total dose of ~32–37 e$^-$ Å$^{-2}$ (dose rate 7–8 e$^-$ px$^{-1}$ s$^{-1}$). The defocus ranged from −2 to −4 μm.

Image processing for the pore form map was performed using Relion2 (version 2.1 beta). A smaller dataset of 157 micrographs was initially subjected to manual particle picking, 2D classification and 3D classification. For the latter, a 30 Å resolution map, created from the TcdA1 pore-form cryo-EM model (PDB:5lkh,5lki) using the molmap command in UCSF Chimera[48], was used as a 3D reference. Five-fold symmetry was clearly evident in 3D reconstructions generated free of symmetry constraints and so in subsequent processing steps C5 symmetry was imposed.

To obtain the final reconstruction, beam-induced motion correction and dose-weighted averaging were performed on 387 micrographs using the Unblur program[46]. CTF estimation was performed using CTFFIND4.1[49]. The 2D classes from the initial processing were used as references for autopicking, which identified ~68,000 particles, extracted with a square box size of 500 pixels. Following 2D class averaging, it was judged that the autopicking had identified a substantial proportion of false negative particles (e.g. carbon edges) with the majority of the true positive particles clustering into 18 classes representing ~30,000 particles. These particles were used for 3D classification, with four 3D classes specified. Particles corresponding to the best 3D class (5130 particles) were then used for auto refinement. The final reconstruction was post-processed using a soft-edged solvent subtraction mask and the resolution of this map was estimated by Fourier shell correlation at the criterion FSC = 0.143.

**Model building and refinement**. Model building into the YenTcA pre-pore map was initiated using a homology model for the translocation pore and the TcB-binding domain, built with SWISS-MODEL[50] using the *Photorhabdus luminsecens* TcdA1 crystal structure as a template (4o9y). Previously determined crystal structures of Chi1 (PDB: 3oa5) and Chi2 (PDB: 4dws) were initially fitted into the map together with the homology model as rigid bodies using SITUS[51] and the fit in map tool in UCSF Chimera. The translocation pore, TcB-binding domain and the crystal structure of Chi2 were refined into the EM map density using ISOLDE[52,53], a plugin enabling interactive, molecular dynamics-guided structure refinement within ChimeraX[54]. The rest of the model was built de novo using ISOLDE and subjected to molecular dynamics-guided structure refinement against the map density, initially for a single protomer and finally for the entire structure with strict five-fold symmetry imposed across the protomers. Model statistics were calculated using the MolProbity server[55]. The Chi1 crystal structure was excluded from the final model due to low resolution of the feet densities, and therefore just rigidly fitted using UCSF Chimera's fit in map (Fig. 2 and Supplementary Fig. 9). The pseudo-atomic model for the YenTcA pore form (Fig. 4) was built using the following domains of the YenTcA pre-pore atomic model, determined in Fig. 1: Chi1 (pink), Chi2 (cyan), pore and TcB-binding domain (yellow/green), neuraminidase-like domain (purple) and a C-terminal region of the A1 shell domain (A1$_{236-1161}$, orange). The domains were initially fitted as protomers using UCSF Chimera's fit in map, then symmetrised after determining the symmetry axis, also in Chimera. Other domains were not included due to ambiguity in the fitting process.

**Structure and sequence analyses**. All visualisation and analysis of atomic structures and 3D volumes were performed in UCSF Chimera[48] or ChimeraX[54]. Electrostatic surface potential calculations were carried out using the Adaptive Poisson-Boltzmann Solver software package (APBS)[56], with a prior PDB2PQR coordinate file preparation according to respective pH[57]. Pore properties were calculated and displayed using the HOLLOW[58] and HOLE[59] tools. Buried surface area of the YenA1 and YenA2 (residues 1–165;559–132) shell domain was calculated using PDBePISA (http://www.ebi.ac.uk/pdbe/pisa/). Sequence alignments for the neuraminidase-like domain (Fig. 3) were generated using the multiple sequence alignment tool T-Coffee and visualised using MView (https://www.ebi.ac.uk/services). Sequence alignments for the pore domain in Supplementary Fig. 5 were generated in T-Coffee and aligned in the sequence manipulation suite[60]. Sequence logo was created with WebLogo[61]. The following Uniprot sequences were utilised for sequence alignments: YenA2 (B6A878_9GAMM), TcdA1 (Q9RN43_PHOLU), TcdA2 (Q8GFA0_PHOLU), TcdA4 (Q8GF92_PHOLU), TcdA2_PA (C7BNH9_PHOAA), XptA1 (D3VHH9_XENNA), XptA2 (D3VHI2_XENNA), SepA (Q9F9Z3_9GAMM), SppA (D5FW91_9GAMM), TcYF1 (Q6XPA4_YERFR).

**Electrophysiology**. Black lipid membranes (BLMs) were prepared by spreading a 1:1 mixture of POPE and POPC solubilised in *n*-decane across a 200 μm diameter hole separating two chambers in the recording apparatus. The buffer contained 10 mM HEPES pH 7.4, 200 mM NaCl. Bilayers with resistances of 1 GΩ were used for recordings. Approximately 0.6–1.2 μg of YenTcA or 2–4 μg of YenTc was added to each chamber and stirred for 30 min at a holding voltage of 20 mV prior to the commencement of recordings. Single channel recordings were filtered at 5 kHz and sampled at 100 μs intervals using HEKA Pulse software v8.8 and analysed using TACX4.3.3.

**Glycan array**. Glycan binding profiles were analysed using a printed array presenting 423 unique glycan structures. Array slides, printed as described previously[62], were pre-blocked for 5 min with PBS supplemented with 2 mM MgCl$_2$, 2 mM CaCl$_2$ and 0.5% BSA (the array buffer). For each YenTc array, 10 μg of protein was incubated (1 h in the dark) with 2 μM Red-NHS dye (Monolith NT™ kit, NanoTemper). For each chitinase array, 1.5 μg of Chi1 or Chi2 was labelled by pre-incubation (5 min in the dark) with a monoclonal mouse anti-His antibody, along with secondary and tertiary Alexafluor 555 rabbit anti-mouse, and goat anti-rabbit antibodies at a ratio of 4:2:1. For each experiment, the array slides were incubated with labelled protein samples at room temperature in the dark (45 min). Following incubation, arrays were washed twice with PBS (the first wash buffer containing 0.5% BSA) and dried by centrifugation at 200 × $g$. Array slides were scanned using an Innopsys Innoscan 1100AL array scanner and the results analysed using ScanArray Express (Perkin Elmer) and Mapix software. Three replicate experiments were performed, with four replicates per experiment ($n = 12$). Binding was only classified as positive if three conditions were met: the response in relative fluorescence units was significantly higher than the background ($P < 0.005$, Student's *t*-test); the response was higher than the adjusted background (defined as the average of negative control spots plus three standard deviations) and the response was positive in all three independent experiments.

**Reporting summary**. Further information on research design is available in the Nature Research Reporting Summary linked to this article.

## Data availability

Data supporting the findings of this manuscript are available from the corresponding author upon reasonable request. A reporting summary for this Article is available as a Supplementary Information file. The electron microscopy maps that were generated during this study are available in the EMDB with the following accession codes: YenTcA pre-pore form EMDB-20053; YenTcA pore form EMDB-20054. The atomic model coordinates generated for the YenTcA pre-pore form are deposited in the PDB under accession code 6ogd. The source data underlying Fig. 5 and Supplementary Fig. 11 are provided as a Source Data file.

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

## Acknowledgements

The authors acknowledge the facilities, and the scientific and technical assistance, of the Australian Microscopy & Microanalysis Research Facility at the Centre for Microscopy and Microanalysis, The University of Queensland, and the BioEM Lab at the University of Basel. We are grateful to Mohamed Chami (University of Basel), Robert McLeod (University of Basel) and Matthias Floetenmeyer (University of Queensland) for their advice and input. This work was supported by grants from the Australian Research Council (DP160101018, DP170104484), the Royal Society of New Zealand Marsden Fund (14-UOA-146) and the New Zealand Foundation of Research, Science and Technology (C10X0804).

## Author contributions

S.J.P., M.J.L., J.S.L., B.H., H.S. and M.R.H.H. designed experiments; S.A.J. and M.R.H.H. provided bacterial strains; S.A.J., J.N.B., S.J.P. and I.C. expressed and purified protein; R.R., L.B., K.N.G. and S.J.P. collected electron microscopy data; S.S., J.K.B. and S.J.P. processed electron microscopy data; T.I.C. and S.J.P. performed model building and model validation; F.S. and J.E.D. performed electrophysiology experiments; I.C., L.H.T. and T.V. designed and performed glycan array experiments; S.J.P. and M.J.L. wrote manuscript; all authors discussed the results, read and edited the manuscript.

## Additional information

**Competing interests:** The authors declare no competing interests.

