## [Peer Review File · Nature Communications]

Reviewers' Comments:

Reviewer #1:

Remarks to the Author:

The manuscript by Piper and colleagues describes a cryoEM structure of large pore-forming toxin from pathogen bacteria *Yersinia entomophaga* belonging to an ABC toxin family. The authors report cryoEM structure of a YenTcA complex at 4.4 Å resolution. The modelled structure allows them to place different TcA proteins (four in total, YenA1, YenA2, Chi1 and Chi2) into the map. In addition they also report low resolution pore structure of YenTcA and this allows to describe molecular events during pore formation. The results thus shed some light onto the pore forming mechanism of *Y. entomophaga* toxins. The authors derive their conclusions based on the comparisons of structures of YenTc in solution and when bound to liposomes, however, they are rarely supported by additional experiments employing mutagenesis and functional assays.

YenTcA belongs to ABC toxins superfamily. The authors have used certain sequential similarities to other ABC toxins in order to build the model, nicely summarized in Supplementary Figure 2. There is less much novel information available, mostly they deal with placement of Chi1 and Chi2 proteins in prepore and pore states. In this respect, the authors should highlight more the novelty of their findings in view of ABC toxins superfamily, or pore forming toxins in general. For example, it is stated in the abstract that ABC toxins combine a conserved pore-forming topology (line 35). It should be specified to which particular pore-forming topology they refer to (I presume it is pore forming topology of *P. Luminescens* ABC toxin).

The structures presented in the paper offer some insights into the regulation of pore forming mechanism. For example pore opening was attributed to an arrangement of five histidines, each from one subunit, in the lumen of the pore. The functional role of these residues (and residue L990, see below) should be confirmed by site-directed mutagenesis, i.e. the replacement of His side chain with a smaller one, that is not dependant on pH, should provide a confirmation for this hypothesis. This is quite an important confirmation, since a lot of reasoning is based on this, i.e. they state that YenTcA pore may have a different gating mechanism (line 164).

Likewise the significant conformational structural rearrangements that are proposed in the paper could be assessed by disulfide scanning mutagenesis to confirm the role of Chi1 in pore formation (i.e. line 216).

It is stressed that residue L990 provides the final closure of the pore (Fig. 3de; line 146), but these panels do not show the position of this residue. I suggest to show it by sticks, similar to the His996 residue in panel 3d.

The authors have also examined glycan binding profile of YenTc in order to understand interactions at the cellular surface by using high-throughput glycan microarray screen. However, no experiment that shows direct binding of chitinase domain with sugars is presented. So no definite conclusions can be reached about interactions of YenTc with carbohydrate receptors at the cell surface. This part of the study should include additional binding experiments by other methods (i.e. surface plasmon resonance) and isolated domains in order to provide more definite answers about the glycolipid receptor and the initial step in the pore forming mechanism.

For sugar-microbial toxin interaction see also Lenarčič et al. (*Science*, 2017) (line 279, references 29-32).

What is the rationale for using POPE:POPC mixture for electrical measurements? Why not just POPC?

Why are there membrane insertions, if there are no glycolipids present that should facilitate membrane binding?

The name of the first author in the first reference is not formatted properly (line 408).

The number of particles used for 3D reconstitutions is modest. Could the resolution be improved by inclusion of more particles?

Reviewer #2:

Remarks to the Author:

Piper et al. describe the cryo-EM structure of component A of the ABC pore forming toxin from *Yersinia entomophaga*. The assembly is composed of four different proteins, YenA1, YenA2, Chi1 and Chi2, which form a supramolecular complex that accounts for pore formation within lipid bilayers of the host membrane. Moreover, the assembly facilitates translocation of the toxin out of the BC-cocoon into the cytosol of the targeted cell.

1. The authors present a 4.4 Å resolution structure of YenTcA in its soluble pre-pore conformation. It is stated that 'Previously we determined a low resolution structure of YenTcA, sufficient to establish its pentameric quaternary structure and proposed an arrangement of structural subunits within it [7]. (lines 64-65)'. Thus, the overall architecture of the toxin has already been reported in 2011.
2. 'Densities corresponding to Chi1 and Chi2 were less well resolved, but models of these regions were obtained by fitting crystal structures of the chitinase domains (PDB IDs 3OA5, 4DWS) [3,15] (lines 78-80)'. Though the structure is presented at 4.4 Å resolution, '[Piper et al.] were unable to locate density corresponding to Chi1 in [their] high resolution cryo-EM map (line 127)' leading to the take home message 'While the orientation of Chi1 within these densities could not be unambiguously determined, rigid fitting (Fig. 2c) indicates that their size is consistent with our previously determined crystal structure of Chi1 [15]'. Thus, I conclude that the data are just conform to the determined X-ray structure determined in 2012, but lack novelty.
3. It was proposed that 'YenTcA pore is likely to respond differently to pH changes than other ABC toxins and may therefore have an altogether different gating mechanism. (lines 163-164)' followed by the argument that 'Indeed, repeated attempts to obtain an alternative conformation of YenTcA by pH titration in vitro have been unsuccessful in our hands. (lines 185-186)'. Therefore, Piper et al. performed 'electrophysiological techniques (Supplementary Fig. 6), similar to those employed previously (lines 186-187)', which have been reported already in 2013. Nevertheless, this approach allowed the authors to obtain a 3D reconstruction of the inserted assembly at 11 Å resolution, which however 'is not sufficient to allow atomic model building (line 193-194)'. In my opinion EM-resolutions of less than 8 Å are too low to get detailed aspects at the molecular level.
4. The last chapter highlights unpublished data 'Genetic knockouts of chi1 or chi2 in *Y. entomophaga* result in a strain that no longer secretes an ABC toxin complex (M. Hurst, unpublished data), leading [Piper et al.] to conclude that Chi1 and Chi2 are essential to form a correctly folded and functional complex. (lines 223-225)'. Moreover it is emphasized that according to a published 'high-throughput glycan microarray screen [25]. (lines 233-234)' ... 'The glycan binding profile of YenTc is similar in apparent complexity to results reported previously for other bacterial pore-forming toxins, including the cholesterol-dependent cytolysin streptolysin O [26]. (lines 237-239)'.
5. Figure 5 displays a cartoon of the proposed mechanism of YenTc pore formation and toxin translocation without any conclusion to the reader: 'The YenTc recognises as yet unidentified receptor(s) on the surface of insect midgut epithelia. (lines 605-606)' followed by 'A conformational change occurs – either prior to, upon or subsequent to receptor recognition (line 607)' and succeeded in 'It remains unclear whether pore formation occurs at the cell surface or within intracellular vesicles

(lines 608-609)'.

In summary I conclude that the submitted manuscript severely lacks novelty and thus, the work is not a suitable candidate to get published in Nature Communications.

Reviewer #3:

Remarks to the Author:

The manuscript presents the structure of YenTcA in a pre and post pore state. Complementary approaches were used to generate the structures of this complex assembly. Of particular interest and impact is the observation of glycan binding activity by YenTc, that may represent a receptor for the toxin.

Comments.

1. YenTcA and YenTc are not obviously and clearly defined in the text. On line 24 we have a definition "The A subunit of the Yersinia entomophaga ABC toxin (YenTcA)". On line 55 we have "secretes an ABC toxin (YenTc)". The nature of YenTcA vs YenTc is only obvious (to me) when looking at the images in panel C of Supplementary figure 6. The authors should provide a more expansive and unambiguous definition of what comprises YenTcA vs YenTc earlier in the manuscript, perhaps using that nice image comparison in panel C of Supplementary figure 6 to support the text.

2. The most impactful aspect of the manuscript is the combination of structure with function. In this case, the functional data presented is the demonstration of glycan binding by YenTc. Unfortunately the data as presented in the manuscript does not have sufficient methodological and supporting primary data to be publishable. Assuming these issues are resolved and the data are assessed as valid, then there is a further issue that the current analysis does not go far enough in two respects. First, in not confirming binding of glycans by independent and quantitative methods, second, by not identifying the subunit of YenTc responsible for the observed binding. I have detailed the issues and suggestions below:

a) Glycan array methodological and supporting primary data issues.

i) It is not sufficient to show a "glycan binding profile" supplementary Figure 8, panel C, without also showing the full dataset that supports the heat map presentation. The statistical analysis used to define what the heat map represents as positive or negative is already described (lines 400-405). The authors must also include a full list of the glycan structures, the results of the assay of YenTc binding to the glycans (the raw fluorescent units or fold difference above controls). Without knowing the structures probed the glycan binding profile provides no information, without the data on the binding to each glycan the result cannot be verified as a valid study.

ii) It is OK to cite previous methods from other publications as the authors do on Line 392 "Glycan binding profiles were analysed using a printed array presenting 423 unique glycan structures. Array slides, printed as described previously" (cites reference 53). However, as noted in (i) above the full set of structures and primary data needs to be presented in this manuscript. Also, the manuscript cited does not describe an array with 423 unique glycans, it describes an array with a lesser number of glycans. Looking at arrays produced by that facility it seems that they need to cite Waespy et al PLoS Negl Trop Dis. 2015, 9:e0004120. doi: 10.1371/journal.pntd.0004120 for the correct, 423 glycan version of the glycan array used in this manuscript.

The authors can use two of the reference they cit as guides fro how to present the complete package with respect to Glycan array data i.e. refs 25, 26

iii) Independent and quantitate methods.

In the field of glycobiology, glycan array studies, like any high throughput semi-quantitate analysis (e.g. expression profile arrays, proteomics, RNA seq), are confirmed by independent quantitate techniques such as isothermal calorimetry or surface plasmon resonance. The glycan array references cited by the authors report these independent methods. These data not only confirm the array data, they provide a hierarchy of affinity that can aid in identification of structure of particular biological significance, in this case candidate toxin receptors.

After reporting, confirming and ranking data (i,ii,iii), a minimal expectation would be some scholarly discussion of the findings in relation to, for example, glycan that may be expected to occur on the epithelium of the mid-gut of insect hosts of *Yersinia entomophaga*.

b) Studies to define the subunit(s) of YenTc responsible for glycan interactions.

Assuming the technical issues with the glycan array data collection, presentation and confirmation are solved, then the main question arises from this study is what part of YenTc is doing the binding? The manuscript directs the reader to expect that Chi1 is the likely candidate:

Line 30: "virulence enhancers Chi1 and Chi2 which occupy sites implicated in host cell recognition. A"

Line 32: "An outward rotation of the Chi1 proteins accompanies membrane insertion, providing access of the protruding translocation pore to the liposomal membrane and we present 33 evidence that this conformational rearrangement may be coupled to receptor binding in vivo."

Line 69 "functional data that suggest the Chi1 and Chi2 proteins are likely to play an important role in membrane binding and cellular recognition by YenTc in vivo." i.e. the glycan array data is the only functional data in the manuscript

Line 210 "approximately 100 Å (Supplementary Video 1). This rearrangement provides space for the exposed pore to insert into a lipid membrane, while maintaining a close proximity of the Chi1 subunits to the membrane surface."

Line 269 "into its structure, Chi1, which we have shown here as being located closest to the membrane surface following membrane bilayer insertion, may instead play a different role in cell surface recognition. While we are yet to identify an explicit receptor for YenTc, a role for Chi1 in this process would be consistent with YenTc binding to a glycan receptor."

Figure 5. Proposed mechanism of YenTc pore formation and toxin translocation - showing Chi1 interacting with a glycan receptor.

After this major focus in the manuscript on Chi1 mediating a potential glycan interaction is disappointing that the authors did not do any studies to address this hypothesis. The lack of data addressing this central hypothesis of the manuscript has a major negative effect on it impact.

To address the hypothesis the authors should attempt some or all of the relatively straightforward studies listed below:

Conduct glycan array studies, and associated quantitative analyses, on Chi1 and compare to YenTc data.

Identify candidate receptors from these studies, and those on YenTc.

Test these receptors in toxin blocking cell assays using free glycans corresponding to the candidate receptors.

Point-by-point response to Reviewer's Comments

Reviewer #1 (Remarks to the Author):

The manuscript by Piper and colleagues describes a cryoEM structure of large pore-forming toxin from pathogen bacteria *Yersinia entomophaga* belonging to an ABC toxin family. The authors report cryoEM structure of a YenTcA complex at 4.4 Å resolution. The modelled structure allows them to place different TcA proteins (four in total, YenA1, YenA2, Chi1 and Chi2) into the map. In addition they also report low resolution pore structure of YenTcA and this allows to describe molecular events during pore formation. The results thus shed some light onto the pore forming mechanism of *Y. entomophaga* toxins. The authors derive their conclusions based on the comparisons of structures of YenTc in solution and when bound to liposomes, however, they are rarely supported by additional experiments employing mutagenesis and functional assays.

We think it is likely that Reviewer 1 has not realised that all of the experiments reported in our paper utilise endogenous protein complexes. The limitation associated with this, is that there is no convenient method for performing site directed mutagenesis, as would be straightforward for toxins produced via recombinant over-expression in for example *E. coli*. Reconstituting complexes of this size and complexity *in vitro* is incredibly challenging technically – we note that while the TcdA1 toxin from *Photobacterium* is able to be produced recombinantly, TcdA1 is a homopentamer whereas the complex studied here contains 4 unique subunits in total and a total of 20 separate protein chains. This highlights the fact that YenTcA is a vastly more complex system, and despite substantial investment of time and effort to produce YenTcA via recombinant methods it has proven intractable. We do believe there are significant attractive advantages to working with endogenous complexes purified directly from the natural host, including that we can be sure that our structures represent the native, secreted state, and not an artefactual structure or complex that arises from production in a foreign organism and/or via reconstitution *in vitro*.

YenTcA belongs to ABC toxins superfamily. The authors have used certain sequential similarities to other ABC toxins in order to build the model, nicely summarized in Supplementary Figure 2. There is less much novel information available, mostly they deal with placement of Chi1 and Chi2 proteins in prepore and pore states. In this respect, the authors should highlight more the novelty of their findings in view of ABC toxins superfamily, or pore forming toxins in general. For example, it is stated in the abstract that ABC toxins combine a conserved pore-forming topology (line 35). It should be specified to which particular pore-forming topology they refer to (I presume it is pore forming topology of *P. luminescens* ABC toxin).

Here we referred to the pore-forming topology of the TcdA1 toxin, as the reviewer presumed. We have clarified this statement in the revised manuscript and also highlight the novelty of the YenTc structure in view of the ABC toxin family. Importantly, our findings help to define an emerging paradigm in pore-forming toxins in general – that subfamilies are defined by conserved pore-forming folds that are decorated with unconserved structures that determine host tropism.

The structures presented in the paper offer some insights into the regulation of pore forming mechanism. For example pore opening was attribute to an arrangement of five histidines, each from one subunit, in the lumen of the pore. The functional role of these residues (and residue L990, see below) should be confirmed by site-directed mutagenesis, i.e. the replacement of His side chain with a smaller one, that is not dependant on pH, should provide a confirmation for this hypothesis. This is quite an important confirmation, since a lot of reasoning is based on this, i.e. they state that YenTcA pore may have a different gating mechanism (line 164).

Likewise the significant conformational structural rearrangements that are proposed in the paper could be assessed by disulfide scanning mutagenesis to confirm the role of Chi1 in pore formation (i.e. line 216).

We agree that these experiments would be desirable under circumstances where site-directed mutagenesis were available to screen for functional mutants, but for the reasons outlined above these experiments are far less than straightforward. For this reason, we chose to focus the majority of our structural interpretation on comparisons to other toxins, with likely mechanisms of activity drawn from these comparisons. We think the claim in our manuscript that YenTcA is likely to have a different gating mechanism is well supported by the evidence we present. Specifically, the unconserved nature of key amino acids known to be involved in gating the TcdA1 pore, our modelling of His996, and the poorly-conserved amino acid sequence of the pore-closing loop of the neuraminidase-like domain which forms an electrostatic lock in the TcdA1 toxin. Based on the amino acid substitutions observed it is chemically impossible for the two structures to be gated in the same way.

It is stressed that residue L990 provides the final closure of the pore (Fig. 3de; line 146), but these panels do not show the position of this residue. I suggest to show it by sticks, similar to the His996 residue in panel 3d.

We have amended the figure as per the suggestion

The authors have also examined glycan binding profile of YenTc in order to understand interactions at the cellular surface by using high-throughput glycan microarray screen. However, no experiment that shows direct binding of chitinase domain with sugars is presented. So no definite conclusions can be reached about interactions of YenTc with carbohydrate receptors at the cell surface. This part of the study should include additional binding experiments by other methods (i.e. surface plasmon resonance) and isolated domains in order to provide more definite answers about the glycolipid receptor and the initial step in the pore forming mechanism.

This point is elaborated by Reviewer 3 and we provide a detailed response in addressing her/his concerns below. While we agree it would be highly desirable to demonstrate quantitative binding, for this particular system there is a major technological barrier that cannot currently be overcome. Quantifying interactions by SPR is entirely dependent on a measurable change in the reflective properties of the chip upon ligand binding – this is usually the result of a major change in molecular weight of the immobilised species. While this approach is common for small protein-ligand interactions, or protein-protein interactions, unfortunately there is no biosensor technique in the world sufficiently sensitive to detect the change that occurs when a sugar molecule of a few hundred Da (at most) binds to a >2 MDa protein complex.

For sugar-microbial toxin interaction see also Lenarčič et al. (Science, 2017) (line 279, references 29-32).

Thank you for the suggestion. We have included the appropriate reference in our revised manuscript.

What is the rationale for using POPE:POPC mixture for electrical measurements? Why not just POPC? Why are there membrane insertions, if there are no glycolipids present that should facilitate membrane binding?

There was no specific reason for using this mixture, other than that it is one commonly used in e-phys studies and reflects the lipid mixtures used where similar studies have been performed on

other bacterial toxins belonging to the ABC family. Regarding the reviewer's second point, again this is an observation that parallels that seen for other ABC toxins. We do note however that in our studies, the rate of incorporation into the membranes is relatively slow (approximately 30 minutes at the concentrations tested), and we would hypothesise that inclusion of a suitable ligand (whether a glycolipid or otherwise) would likely accelerate the rate of incorporation in vivo. We would be happy to speculate on this in the manuscript if it is deemed suitable to do so.

The name of the first author in the first reference is not formatted properly (line 408).

The formatting, while unusual, is actually correct here.

The number of particles used for 3D reconstructions is modest. Could the resolution be improved by inclusion of more particles?

In short, no (or at best, minimally).

In the time since original submission, we have evaluated the effect of dataset size on our reconstructions. Increasing the dataset size to as many as 60,000 achieves only a modest increase in average resolution to 4.2Å when comparing with equivalent masks. Similarly, reducing the particle set size to 5,000 particles does not significantly reduce the resolution estimate. We attribute this to the fact that the peripheral domains of YenTcA appear to be very mobile with no fixed location, and thus limit the achievable average resolution. It is worth pointing out in this regard that the highest resolution cryo-EM structures currently being reported are of very rigid structures that exhibit low or no conformational dynamics. 4Å is not unusual for a dynamics assembly and many of the reported structures that appear to exceed this actually achieve inflated resolution estimates by simply masking out the most rigid parts of the structure. We could (and have) obtained nominally "higher resolution" structures simply by masking out more of the poorly resolved peripheral densities, but this simply removes these details from the map and does not substantially improve the local resolution of the more well defined regions. The fudging of the resolution estimate in this way does absolutely nothing to improve or enhance the biological insights provided by the structure and we share the opinion of many other structural biologists who believe that more important than the resolution "number" is the structural insights that the map actually provides. We believe strongly that our map at its current resolution provides substantial new structural insights that are commensurate with the estimated resolution.

Reviewer #2 (Remarks to the Author):

Piper et al. describe the cryo-EM structure of component A of the ABC pore forming toxin from *Yersinia entomophaga*. The assembly is composed of four different proteins, YenA1, YenA2, Chi1 and Chi2, which form a supramolecular complex that accounts for pore formation within lipid bilayers of the host membrane. Moreover, the assembly facilitates translocation of the toxin out of the BC-cocoon into the cytosol of the targeted cell.

1. The authors present a 4.4 Å resolution structure of YenTcA in its soluble pre-pore conformation. It is stated that 'Previously we determined a low resolution structure of YenTcA, sufficient to establish its pentameric quaternary structure and proposed an arrangement of structural subunits within it [7]. (lines 64-65)'. Thus, the overall architecture of the toxin has already been reported in 2011.

This statement is factually incorrect. In our 2011 paper we determined a low resolution structure (18 Å in negative stain). This led us to *propose* a subunit arrangement, but one of the many significant advances reported in the current manuscript is that the Chi1 subunit was mislocated in this 2011 structure (due to its low resolution). Furthermore, the conclusions that can be drawn from a 4Å map are profoundly more significant in the context of molecular mechanism and structure than those that can be drawn from an 18Å map.

2. ‘Densities corresponding to Chi1 and Chi2 were less well resolved, but models of these regions were obtained by fitting crystal structures of the chitinase domains (PDB IDs 3OA5, 4DWS) [3,15] (lines 78-80)’. Though the structure is presented at 4.4 Å resolution, ‘[Piper et al.] were unable to locate density corresponding to Chi1 in [their] high resolution cryo-EM map (line 127)’ leading to the take home message ‘While the orientation of Chi1 within these densities could not be unambiguously determined, rigid fitting (Fig. 2c) indicates that their size is consistent with our previously determined crystal structure of Chi1 [15].’ Thus, I conclude that the data are just conform to the determined X-ray structure determined in 2012, but lack novelty.

We state in our manuscript that while the Chi1 crystal structure was fitted as a rigid body only, the Chi2 crystal structure was fitted into the map density initially, and then refined using the tools in ISOLDE (as were the A1 and A2 subunits refined, followed by manual building of the structures of these subunits). Concerning the chitinases specifically, there are three major outcomes: 1) we were able to identify the correct location of Chi1; 2) we were able to unequivocally identify that Chi2 (and not Chi1) is located at the lateral periphery of the complex; 3) we were able to remodel parts of Chi2, reflecting differences in the crystal structure and complex-bound conformations. Moreover, there is no existing structure of either the YenA1 or YenA2 subunits, which represent ~70% of the complex by molecular weight. Thus the determination of structures for YenA1 and YenA2 is an entirely novel finding not previously reported. The statement that our cryoEM structure is “just conform” to previously determined crystal structures is factually incorrect.

3. It was proposed that ‘YenTcA pore is likely to respond differently to pH changes than other ABC toxins and may therefore have an altogether different gating mechanism. (lines 163-164)’ followed by the argument that ‘Indeed, repeated attempts to obtain an alternative conformation of YenTcA by pH titration in vitro have been unsuccessful in our hands. (lines 185-186)’. Therefore, Piper et al. performed ‘electrophysiological techniques (Supplementary Fig. 6), similar to those employed previously (lines 186-187)’, which have been reported already in 2013. Nevertheless, this approach allowed the authors to obtain a 3D reconstruction of the inserted assembly at 11 Å resolution, which however ‘is not sufficient to allow atomic model building (line 193-194)’. In my opinion EM-resolutions of less than 8 Å are too low to get detailed aspects at the molecular level.

We limited our interpretation of the pore structure to the level of interpreting re-positioning of individual subunits, entirely appropriate for a structure at this resolution. It is remarkably clear that there are significant conformational changes between the pre-pore and pore configuration clearly visualised at this resolution. In support of this statement, we have now modelled subdomains of the pre-pore structure into our liposome-embedded reconstruction as rigid bodies. These domains can be unambiguously fitted into the map and identify clear structural rearrangements that are unequivocally resolved at the current resolution and informative of functional mechanisms.

Moreover, our interpretation of molecular mechanisms is not limited to this 11 Å map, rather it is drawn from all of the data presented in our paper which includes the high resolution EM structure in the pre-pore configuration, as well as sequence comparisons with TcdA1, where key residues involved in gating the latter are not conserved in YenTcA.

4. The last chapter highlights unpublished data ‘Genetic knockouts of chi1 or chi2 in *Y. entomophaga* result in a strain that no longer secretes an ABC toxin complex (M. Hurst, unpublished data), leading [Piper et al.] to conclude that Chi1 and Chi2 are essential to form a correctly folded and functional complex. (lines 223-225)’. Moreover it is emphasized that according to a published ‘high-throughput glycan microarray screen [25]. (lines 233-234)’ ... ‘The glycan binding profile of YenTc is similar in apparent complexity to results reported previously for other bacterial pore-forming toxins, including the cholesterol-dependent cytolysin streptolysin O [26]. (lines 237-239)’.

It is not clear to us why the Reviewer has reproduced these statements. To be clear, we conclude that the apparent complexity of the glycan binding profile is similar to other toxins, but not the specific glycans that it recognises on the array. We reiterate that this piece of evidence represents the first attempt to identify a receptor for any ABC toxin and is therefore highly novel, and we have included an expanded glycan array analysis in our revised manuscript (detailed response to Reviewer 3 below). Regardless, the major novel finding of our paper lies in the structural characterisation of YenTcA and it is unfortunate that this was not recognised by the reviewer.

5. Figure 5 displays a cartoon of the proposed mechanism of YenTc pore formation and toxin translocation without any conclusion to the reader: ‘The YenTc recognises as yet unidentified receptor(s) on the surface of insect midgut epithelia. (lines 605-606)’ followed by ‘A conformational change occurs – either prior to, upon or subsequent to receptor recognition (line 607)’ and succeeded in ‘It remains unclear whether pore formation occurs at the cell surface or within intracellular vesicles (lines 608-609)’.

This figure was intended to summarise the current state of understanding of the mechanism associated with ABC toxins. Given that the reviewer has failed to see the novelty of our findings, it is not surprising that they consider the cartoon to be uninformative. We disagree, but would agree to remove the figure (figure 6 in revised manuscript) or move it to supplementary materials if it were ultimately considered unnecessary at an editorial level.

In summary I conclude that the submitted manuscript severely lacks novelty and thus, the work is not a suitable candidate to get published in Nature Communications.

As is clear from the responses above, we disagree in the strongest possible terms. We feel very strongly that there is significant novelty associated with our structure. **i)** YenTc is the first ABC toxin with a “split A” architecture to have its structure determined; **ii)** The pore-forming apparatus has a similar overall fold to the previously characterised TcdA1 toxin but key residues implicated in gating of the pore and conformational change are not conserved, suggesting their mechanisms of activation are likely to be different; **iii)** Moreover, the extent of this similarity in overall fold was not obvious or predicted from sequence comparisons and is only now apparent due to the high resolution structure we report here; **iv)** Motifs most likely to be involved in receptor recognition are also not conserved, which explains the unique host specificity profiles associated with ABC toxins; **v)** We identify unequivocally the location of two chitinase subunits (previous efforts were hampered by the low resolution of our earlier structure – Landsberg, PNAS, 2011) - the identification of chitinases in association with an ABC toxin (or any pore-forming toxin) has never been reported previously.

Reviewer #3 (Remarks to the Author):

The manuscript presents the structure of YenTcA in a pre and post pore state. Complementary approaches were used to generate the structures of this complex assembly. Of particular interest

and impact is the observation of glycan binding activity by YenTc, that may represent a receptor for the toxin.

Comments.

1. YenTcA and YenTc are not obviously and clearly defined in the text. On line 24 we have a definition "The A subunit of the Yersinia entomophaga ABC toxin (YenTcA)". On line 55 we have "secretes an ABC toxin (YenTc)". The nature of YenTcA vs YenTc is only obvious (to me) when looking at the images in panel C of Supplementary figure 6. The authors should provide a more expansive and unambiguous definition of what comprises YenTcA vs YenTc earlier in the manuscript, perhaps using that nice image comparison in panel C of Supplementary figure 6 to support the text.

We added a better explanation of YenTc vs YenTcA in the abstract and also in the introduction. We also incorporated the cartoon from Supp Fig 6 into Figure 1 instead to establish the difference earlier.

2. The most impactful aspect of the manuscript is the combination of structure with function. In this case, the functional data presented is the demonstration of glycan binding by YenTc. Unfortunately the data as presented in the manuscript does not have sufficient methodological and supporting primary data to be publishable. Assuming these issues are resolved and the data are assessed as valid, then there is a further issue that the current analysis does not go far enough in two respects. First, in not confirming binding of glycans by independent and quantitative methods, second, by not identifying the subunit of YenTc responsible for the observed binding. I have detailed the issues and suggestions below:

a) Glycan array methodological and supporting primary data issues.

i) It is not sufficient to show a "glycan binding profile" supplementary Figure 8, panel C, without also showing the full dataset that supports the heat map presentation. The statistical analysis used to define what the heat map represents as positive or negative is already described (lines 400-405). The authors must also include a full list of the glycan structures, the results of the assay of YenTc binding to the glycans (the raw fluorescent units or fold difference above controls). Without knowing the structures probed the glycan binding profile provides no information, without the data on the binding to each glycan the result cannot be verified as a valid study.

We have added the requested material

ii) It is OK to cite previous methods from other publications as the authors do on Line 392 "Glycan binding profiles were analysed using a printed array presenting 423 unique glycan structures. Array slides, printed as described previously" (cites reference 53). However, as noted in (i) above the full set of structures and primary data needs to be presented in this manuscript. Also, the manuscript cited does not describe an array with 423 unique glycans, it describes an array with a lesser number of glycans. Looking at arrays produced by that facility it seems that they need to cite Waespy et al PLoS Negl Trop Dis. 2015, 9:e0004120. doi: 10.1371/journal.pntd.0004120 for the correct, 423 glycan version of the glycan array used in this manuscript.

The authors can use two of the reference they cite as guides for how to present the complete package with respect to Glycan array data i.e. refs 25, 26

We thank the reviewer for pointing this out and have corrected the references.

iii) Independent and quantitative methods.

In the field of glycobiology, glycan array studies, like any high throughput semi-quantitative analysis (e.g. expression profile arrays, proteomics, RNA seq), are confirmed by independent quantitative techniques such as isothermal calorimetry or surface plasmon resonance. The glycan array references cited by the authors report these independent methods. These data not only confirm the array data, they provide a hierarchy of affinity that can aid in identification of structure of particular biological significance, in this case candidate toxin receptors.

We appreciate the useful comments of the reviewer in this regard and have also been careful not to over-interpret the results of our semi-quantitative analysis in the context of their *in vivo* significance. As we outlined above, while surface plasmon resonance is widely used to validate hits from glycan array screens, this is not feasible for the system described here as the increase in molecular weight following binding of a small glycan ($\sim 10^2$ Da) to the very large YenTc ($\sim 10^6$ Da) is on the order of 0.01%. Modern biosensors/SPR are not sufficiently sensitive to detect such a change. While ITC is a suitable alternative, the amount of protein required for these experiments is not currently feasible in the absence of a recombinant expression system for YenTc. We reiterate that all of our experiments have been performed using proteins isolated from the source bacterium and have highlighted this point in our resubmitted manuscript.

After reporting, confirming and ranking data (i,ii,iii), a minimal expectation would be some scholarly discussion of the findings in relation to, for example, glycan that may be expected to occur on the epithelium of the mid-gut of insect hosts of *Yersinia entomophaga*.

A more comprehensive discussion of the physiological significance of classes of glycans identified in the screens has been included in the revised manuscript.

b) Studies to define the subunit(s) of YenTc responsible for glycan interactions.

Assuming the technical issues with the glycan array data collection, presentation and confirmation are solved, then the main question arises from this study is what part of YenTc is doing the binding? The manuscript directs the reader to expect that Chi1 is the likely candidate:

Line 30: "virulence enhancers Chi1 and Chi2 which occupy sites implicated in host cell recognition. A"

Line 32: "An outward rotation of the Chi1 proteins accompanies membrane insertion, providing access of the protruding translocation pore to the liposomal membrane and we present 33 evidence that this conformational rearrangement may be coupled to receptor binding *in vivo*."

Line 69 "functional data that suggest the Chi1 and Chi2 proteins are likely to play an important role in membrane binding and cellular recognition by YenTc *in vivo*." i.e. the glycan array data is the only functional data in the manuscript

Line 210 "approximately 100 Å (Supplementary Video 1). This rearrangement provides space for the exposed pore to insert into a lipid membrane, while maintaining a close proximity of the Chi1 subunits to the membrane surface."

Line 269 "into its structure, Chi1, which we have shown here as being located closest to the membrane surface following membrane bilayer insertion, may instead play a different role in cell surface recognition. While we are yet to identify an explicit receptor for YenTc, a role for Chi1 in this

process would be consistent with YenTc binding to a glycan receptor."

Figure 5. Proposed mechanism of YenTc pore formation and toxin translocation - showing Chi1 interacting with a glycan receptor.

After this major focus in the manuscript on Chi1 mediating a potential glycan interaction is disappointing that the authors did not do any studies to address this hypothesis. The lack of data addressing this central hypothesis of the manuscript has a major negative effect on it impact.

To address the hypothesis the authors should attempt some or all of the relatively straightforward studies listed below:

Conduct glycan array studies, and associated quantitative analyses, on Chi1 and compare to YenTc data.

Identify candidate receptors from these studies, and those on YenTc.

Test these receptors in toxin blocking cell assays using free glycans corresponding to the candidate receptors.

We thank the reviewer again for these helpful suggestions and have taken these into account in our revised manuscript. Again it is important to point out that recombinant expression of YenTc has to date not been reported or established and so many of the suggested experiments which might be considered straightforward for recombinant proteins are incredibly difficult to perform with native proteins. Site-directed mutagenesis is not straightforward nor well established in *Y. entomophaga* and our only effort to produce a deletion strain (one lacking *chi1* and *chi2*) resulted in the complex no longer being secreted (possibly not assembled). We have however replicated our studies on YenTc using isolated Chi1 and Chi2 subunits and examined the hits comparatively across all three screens. While some commonalities were identified that might support the hypothesis that Chi1 is involved in receptor binding, we also cannot rule out the possibility at this stage that YenTc contains other lectin domains which - either in isolation or in concert with Chi1 - may also contribute to receptor binding. We have substantially redrafted the relevant sections of the paper to 1) discuss this new data; 2) soften the pre-existing hypothesis that Chi1 is a mediator of cell surface recognition; 3) highlight the barriers to further validation; and 4) surmise that regardless, these studies provide an important piece of evidence that, in the absence of any other data on receptor recognition on ABC toxins, our studies provide an important first piece of evidence that glycans may play an important role, as is the case for many other bacterial toxins.

Regarding this last point, both Reviewer 1 and 3 agreed with us that the finding that the receptor for YenTc may be glycan in nature is significant and interesting— we emphasise that there has been *no attempt reported previously to identify a candidate cell surface receptor for this or any ABC toxin*, thus this aspect of the work is highly novel. Nonetheless, we feel that the most interesting and important aspect of the work we present here is the structural characterisation of YenTcA, which establishes against the background of only a *single other example* of an ABC toxin structure, that these bacterial toxins have diverse structures and the underlying assumption based on current literature – that all ABC toxins look and function like TcdA1 – is clearly not the case. Thus, we believe the combined novelty of our structural data and functional insights offered by our semi-quantitative receptor binding studies, we believe, justify publication in *Nature Communications*.

Reviewers' Comments:

Reviewer #1:

Remarks to the Author:

The manuscript was changed considerably in the revised version and is now emphasizing much better the novel findings by the presented structural work and highlighting the differences and major improvements from the previous work, i.e. in the abstract, in the introduction and in the discussion. Structural work is done carefully and in my opinion appropriately argued in the response with regards to the novelty of the presented work.

However, I still have problems with the functional part of the paper. The authors maintain to include glycan binding data and actually provide some new data on glycan arrays for Chi1 and Chi2 proteins (Figure 5 and Supp figure 11). And a significant part of the results section, at the very end, is now devoted to discussion about specificity against different sugars and their presence in different hosts. However, glycan data should be confirmed independently by SPR assays. I do not agree that SPR studies are not feasible with small ligands. There is plenty of SPR literature with modern systems that allow obtaining reliable data for low molecular weight substances, such as sugars in this manuscript. The authors shall perform these experiments, since affinity constants towards different sugars may help interpreting the binding mechanism and also better discuss host tropism. Furthermore, the data for Chi2 presented in supp figure 11, in my opinion, are the most interesting, since there are two sugars (or more?) that show the biggest response (app 12 RFU as opposed to app. 1 for the background and 3-4 for some of the other hits in other proteins). These sugars should be appropriately identified in the manuscript and supportive SPR experiments performed (providing these sugars are available commercially).

As per suggestions of authors, I propose to include a brief section explaining behaviour of the pore formation (slow incorporation in membranes) at the suboptimal lipid membrane composition used in planar lipid bilayers to assess functionality. I think this is actually very important result as it shows that their preparation is functional and I propose more text is devoted to describe the properties of the pores (lines 230-233).

Reviewer #3:

Remarks to the Author:

The authors have fully addressed all of my concerns in the revised manuscripts.